



# Identifying meteorological influences on marine low cloud mesoscale morphology using deep learning classifications

Johannes Mohrmann[1], Robert Wood[1], Tianle Yuan[2,3], Hua Song[4], Ryan Eastman[1], Lazaros Oreopoulos[2]

[1]Department of Atmospheric Sciences, University of Washington, Seattle, WA, USA
[2]Earth Science Division, NASA Goddard Space Flight Center, Goddard, MD, USA
[3]Joint Center for Earth Systems Technology, University of Maryland, Baltimore County, Baltimore, MD, USA
[4]Science Systems and Application, Inc., Lanham, MD, USA

*Correspondence to*: Johannes Mohrmann (jkcm@uw.edu)

**Abstract.** Marine low cloud mesoscale morphology in the southeastern Pacific Ocean is analyzed using a large dataset of
machine-learning generated classifications spanning three years. Meteorological variables and cloud properties are composited
by mesoscale cloud type, showing distinct meteorological regimes of marine low cloud organization from the tropics to the
midlatitudes. The presentation of mesoscale cellular convection, with respect to geographic distribution, boundary layer
structure, and large-scale environmental conditions, agrees with prior knowledge. Two tropical and subtropical cumuliform
boundary layer regimes, suppressed cumulus and clustered cumulus, are studied in detail. The patterns in precipitation,
circulation, column water vapor, and cloudiness are consistent with the representation of marine shallow mesoscale convective
self-aggregation by large eddy simulations of the boundary layer. Although they occur under similar large-scale conditions,
the suppressed and clustered low cloud types are found to be well-separated by variables associated with low-level mesoscale
circulation, with surface wind divergence being the clearest discriminator between them, whether reanalysis or satellite
observations are used. Clustered regimes are associated with surface convergence and suppressed regimes are associated with
surface divergence.

## 1 Introduction

Marine low clouds are radiatively important, with a strong cooling effect on the planet. They also display a wide
range of morphologies, which have differing radiative properties (Chen et al., 2000). Classically, ship-based observations have
classified marine low clouds using the World Meteorological Organization (WMO) cloud types that many are familiar with
(stratocumulus (Sc), cumulus (Cu), etc.) (e.g. Warren et al., 1988). However, clouds also form larger mesoscale
morphologically distinct organizations that would not be apparent from the limited perspective of a surface-based observer.
These mesoscale cloud patterns are of particular interest for two main reasons. First, they have been shown to represent
different underlying marine boundary layer (MBL) regimes (e.g. Wood and Hartmann, 2006; hereafter WH06), namely the
presence of some additional environmental property of the MBL that covaries with cloud morphology. Second, larger
mesoscale patterns are clearly visible from current-generation satellite imagers, allowing for their classification and the
subsequent generation of a potentially informative dataset on a near-global and highly temporally resolved scale. Prior work



has also shown that the relationship between albedo and cloud fraction varies with mesoscale organization (McCoy et al., 2017). Combined with advances in computer image recognition, this creates potential for effective global-scale observational study of MBL clouds and their drivers.

35       In the midlatitude storm tracks and eastern ocean subtropical Sc decks, stratiform low cloud types dominate (Hartmann et al., 1992). These high cloud fraction cloud types are particularly effective coolers, and as a result their organization and structure are the subject of much investigation (Agee, 1987; Muhlbauer et al., 2014). In lower latitudes and away from the eastern subtropical ocean basins, Sc clouds are rarer, and instead we often find boundary layers dominated by cumuliform cloud types, sometimes clustering into large convectively active regions, and sometimes in relatively isolated

smaller Cu. Figure 1, adapted from an observation-based climatic cloud atlas (Hahn and Warren, 2007), shows the difference between the frequency of occurrence of Cu clouds and that of Sc clouds; the commonly-occurring 'Cu-under-Sc' case is classified as Sc as that is consistent with the view from above (Hahn et al., 2001). Red values indicate more Cu and show that boundary layer clouds over the ocean between 30˚N and S are more often cumuliform. Although the average cloud radiative effect (CRE) of these clouds is lower, their ubiquity combined with a high mesoscale variability in cloud fraction makes them

an important target of study.

      Cumuliform MBLs have been observed to contain mesoscale aggregates of shallow convection in a number of different forms (LeMone and Meitin, 1984; Nicholls and Young, 2007). Bretherton and Blossey (2017) (hereafter BB17) demonstrated how mesoscale aggregation of warm shallow Cu presents in Large Eddy Simulation (LES). In their conceptual model, the shallow convective self-aggregation is driven by convection-circulation-humidity feedbacks. These result in cloudy

regions of aggregated convection with a positive mesoscale column water vapor and moisture anomaly, as well as a strong low-level circulation with lower boundary layer convergence acting to further concentrate moisture into the moist columns. The difference between this and the conceptual model for deep-convective self-aggregation (e.g. Emanuel et al. 2014) is that the latter relies on radiative feedbacks, while these are not necessary to produce shallow mesoscale aggregation. BB17 demonstrated that the presentation of shallow aggregation agrees with this conceptual model and suggested that further

observational validation is warranted.

      When classifying stratocumulus and cumulus clouds, a common form of mesoscale variability is mesoscale cellular convection (MCC) (Agee, 1987). This can take the form of open-cellular or closed-cellular MCC. WH06 used a neural network to classify low cloud scenes from satellite observations over the eastern subtropical Pacific Ocean into four categories, based on their MCC type: open, closed, cellular but disorganized, and no MCC present. The utility of these classification-based

approaches is evident in their ability to show the controls on cloud morphology in cold air outbreaks (McCoy et al., 2017), characterize properties and occurrences of the underlying regimes (Muhlbauer et al., 2014), or discern whether mesoscale morphology is more strongly driven by internal mechanisms or by large-scale meteorology (WH06). However, a limitation of the WH06 classification scheme is its lack of discrimination between cloud morphologies over the warmer regions of the Tropical trades, where MCC is less dominant. Additionally, the power spectra- and Fourier transform-based feature vectors

used for classification were very sensitive to the presence of high cloud, necessitating a strict exclusion of many otherwise



visually-identifiable scenes. More recent investigation of low-latitude marine low cloud mesoscale variability agnostic to previously-identified forms of organization has also been successful in identifying distinct morphological regimes, using machine learning to classify a large dataset of cloud images (Stevens et al., 2020).

In this work we continue the exploration of marine low cloud morphology, its drivers and characteristics, with a new classification scheme first presented in Yuan et al. (2020), and expanding on WH06. The new scheme focuses on discrimination between different cumulus-dominated cloud types, particularly in the Tropical trade wind regions. The machine learning approach taken in this new dataset uses convolutional neural nets (CNN) to permit the inclusion of some scenes with thin or small amounts of high cloud. We also add two cumuliform low cloud morphological types, clustered convection and suppressed convection, to capture more cloud morphological variability in the tropics and subtopics. We start with a description

of the new classification scheme and observational datasets used (section 2) and present the physical characteristics of the resulting cloud types (section 3). Specifically, in section 3 we validate whether the presentation of the two cumuliform cloud types is consistent with the model for mesoscale aggregation of shallow cumulus convection described by BB17. We conclude with a discussion of the significance of these results (section 4).

## 2 Datasets and Methods

Most analysis in this work consists of composite analysis of various observational and model datasets by cloud type. We first describe the cloud type classifications, then the datasets used, and finally the compositing methodology.

### 2. 1 Cloud type classifications

     The classification dataset used is derived from imagery by the Moderate-Resolution Imaging Spectrometer (MODIS), aboard the Aqua satellite. MODIS RGB visible imagery of 128x128 km$^2$ (approximately 1°x1°) cloudy scenes, filtered to

remove scenes with >10% coverage of high cloud, low cloud <5%, and viewing angles >45°, are manually classified as being either comprised mostly of stratus cloud, closed-cellular marine cellular convective Sc (closed MCC), open-cellular Sc (open MCC), disorganized cellular stratocumulus (disorganized MCC), clustered cumulus, and suppressed cumulus. These scenes are then used to train a convolutional neural net, which in turn is run near-globally on such MODIS oceanic scenes. Examples of these types can be found in Figure 2, and a detailed description of the classification dataset and training can be found in

Yuan et al. (2020). For this paper, most analysis is based on three years of classifications from the southeast Pacific (SEP) region, (65°S-equator, 140°W-40°W) which includes much of the Southern Ocean and a small portion of the southwest Atlantic, as well as classifications from summer 2015 in the northeast Pacific (NEP) region (equator-60°N, 180°W-100°W) for co-location with aircraft data (see Section 3.5 below). The resulting tabular dataset contains location, time, and cloud scene classification, as well as MODIS low cloud fraction derived from the MODIS Cloud Product cloud top heights (MYD06,

Platnick et al. 2017). There were approximately 750,000 scenes for the SEP (averaging approximately 65 classifications per MODIS granule and 11 granules per day), while the NEP dataset is smaller with ~35,000 data points. Relative distributions





for the various cloud scene types are provided in Figure 3. Note that these are relative distributions only, normalized for each location. Geographic differences in cloud cover and satellite sampling, the number of viable scenes is not distributed evenly over the region of interest, with approximately 5 time as many scenes in the subtropical Sc region as in the midlatitudes.

**2.2 Satellite-derived ancillary data**

Surface wind divergence is derived from the Advanced SCATterometer (ASCAT) aboard MetOp-A, specifically the 0.25° gridded wind vectors (Ricciardulli and Wentz, 2016). For each classification datapoint, the 1°x1° co-located calculated ASCAT divergence values are extracted and averaged together. Since the ASCAT swath width is much narrower than that of MODIS (even when filtering out high viewing angle scenes), some classified scenes (approximately 45%) had no wind data.

Additionally, the overpass time of MetOp-A (~9:30 a.m. local time) does not coincide with Aqua (~1:30 p.m. local time), and so any significant diurnal cycle in wind divergence could influence results. While this is a source of noise and a point of potential improvement for future work, the diurnal amplitude in surface divergence is likely much smaller than the mesoscale variations (Wood et al., 2009) and so we do not expect any significant biases. We repeat all ASCAT-based analysis with reanalysis wind data as well, which is temporally better-matched, with similar results.

Column water vapor (CWV) is provided by the Advanced Microwave Sounding Radiometer (AMSR-2) aboard the Global Change Observation Mission (GCOM-W1), again using 0.25° gridded daily observations (Wentz et al., 2014). Being on the A-Train as Aqua, GCOM-W1 overpass times are roughly 1:30 p.m. local time, which is nearly simultaneous with the MODIS observations.

For retrieving rain rates, a precipitation dataset based on AMSR-2 89 GHz brightness temperatures and CloudSat

observations is used (Eastman et al, 2019). This particular dataset has the advantage of being calibrated specifically for warm rain from shallow marine clouds, with greater sensitivity to light rain than other passive microwave rain products (Eastman et al., 2019).

To assess the radiative impacts of our cloud types, we also analyze data from the Clouds and the Earth's Radiant Energy System (CERES), specifically SYN1deg hourly data, providing 1° top-of-atmosphere (TOA) all-sky and clear-sky

longwave (LW) and shortwave (SW) fluxes. These are also co-located with the classified cloud scenes both spatially and temporally, and used to calculate the LW, SW, and total cloud radiative effect (CRE) for each classified scene (here F is an upward flux):

$$CRE_{LW} = F_{LW,clear} - F_{LW,all} \qquad CRE_{SW} = F_{SW,clear} - F_{SW,all} \qquad CRE_{tot} = CRE_{LW} + CRE_{SW} \quad 1A$$

**2.3 Reanalysis data**

For the purpose of analyzing large-scale meteorology, as well as comparing to satellite observations, data from the Modern-Era Retrospective analysis for Research and Application, Version 2 (MERRA2) was included in our analysis. The data used was at a 3-hourly resolution, with the time nearest the MODIS overpass selected. In addition to available variables (sea surface temperature, near-surface winds), we derive the estimated inversion strength (EIS) following Wood and Bretherton



(2006), a surface divergence estimate from the 10m winds, and a large-scale divergence $D$ estimate from the 700 hPa heights

and vertical motion from:

$$D_{700} = \frac{w_{700}}{z_{700}} \qquad w_{700} \approx -\frac{\omega_{700}}{\rho_{700}g}$$

Note that this large-scale divergence is not the horizontal divergence at 700 hPa, but rather the mean divergence from the surface to the 700 hPa. The terms large-scale divergence and subsidence are used interchangeably throughout; divergence is plotted instead of subsidence to allow for better comparison with surface divergence. As surface pressure varies with time, the

second equality is only approximate.

For all of the above variables (from reanalysis and satellite), and for each MODIS scene for which we have a classification, we extract the variable in a 1°x1° box centered on the cloud scene to calculate a mesoscale average value, and use the mean over a 10°x10° box for the synoptic mean value. These can then be used together to calculate a mesoscale perturbation, which is simply the difference between the 1°x1° and 10°x10° averages. We also calculate a climatological 1°x1°

average, by using seasonally averaged data.

## 2.4 Aircraft observations

To provide insight into the vertical structure of the boundary layer as well as in-situ cloud observations we use aircraft observations from the Cloud System Evolution in the Trades (CSET) field campaign (Albrecht et al., 2019), which took place in summer 2015. This campaign is particularly suitable since it contained a large number of aircraft profiles and dropsondes

throughout the depth of the marine boundary layer, on a transect spanning from California to Hawaii, and therefore sampling from the Sc-dominated near-coastal region (where organized MCC frequently is found) through the Sc-Cu transition, to the cumuliform tropical MBL. All cloud types other than midlatitude Sc were therefore sampled. The campaign profiles allowed us to estimate the boundary layer depth and degree of decoupling following Mohrmann et al. (2019), and composite by cloud type.

## 2.5 Data compositing by cloud type


Much of the following analysis is of the type shown in figures such as Figure 6, which shows the net cloud radiative effect (CRE) for each cloud type (for the SEP region). For this figure, the ~750,000 classifications are split by year and then further split by scene type. The mean net CRE for each year and type is then plotted. The large sample size makes the sampling uncertainty negligible (error bars representing the standard error of the mean are plotted throughout, though are typically not

visible). This is true even after accounting for the high autocorrelation in the data. The data is nevertheless split by year to demonstrate robustness as represented by (low) interannual variability.

An issue with the observational data compositing approach is that the cloud types do not all have the same geographic distribution. One approach would be to try to coerce geographic parity by sampling the same number of points from some grid, or else to control for every other variable by stratifying the data in many dimensions. Here we take a different approach: to





identify the extent to which differences in potential driver variables reflect short-lived anomalies as opposed to geographic
       sampling bias, we calculate seasonal climatologies for each gridded dataset, and then extract for each scene the climatological
       value of that field at that location. These are also composited by cloud scene type and compared to the composite of
       instantaneous values. This analysis is similar to the mesoscale-vs-synoptic mean comparison described in the previous section,
       but now showing temporal deviations from local climatology. Figure 7 (a) shows all three averages on the same panel for
direct comparison. The black circles represent, averaged over every classification in the dataset, the mesoscale (i.e. 1°x1°
       average) SST at that location and time, the black diamonds are the same but averaged over a 10°x10° box, and the black
       squares again a 1°x1° average, but with seasonal averages instead of daily values of SST.

## 3 Results

### 3.1 Climatology of occurrence

170        We first present the characteristics of the cloud types represented by the classifier categories. This is in addition to
       the analysis presented in Yuan et al. (2020), which presents example scenes, cloud optical thickness, droplet effective radius,
       and absolute frequency for each cloud type. Figure 3 shows the relative frequency of occurrence of the six cloud types in the
       classification scheme. The more stratiform MCC types (a-d) occur at higher latitudes and towards the eastern SEP basin, while
       the two cumuliform types (e and f) dominate the warmer (sub) tropical oceans away from the continents, consistent with the
ship-based climatology of Figure 1. The location of the MCC types (with closed-cell upwind, open-cell downwind) is mostly
       consistent with their occurrence in the WH06 classifications. Both subtropical and midlatitude MCC are identified. The main
       differences with the WH06 classifications are that the disorganized MCC type, which previously included all scenes not
       classified as open MCC, closed MCC, or stratus, now primarily occurs near the Sc cloud deck, instead of spreading over a
       much larger region. Another significant departure is that open-cellular MCC occurs much less frequently than in the WH06
classifier, representing only 4% of all scenes. The solid stratus type is a mix of coastal stratus and midlatitude frontal stratus.
           An ideal cloud type classification scheme would produce useful discrimination between cloud types in all regions, as
       opposed to having one cloud type completely dominate one region and another completely dominate another region. One way
       to visualize how well this classification scheme embodies this property is by considering, for each region, the fraction of all
       scenes which come from the most common cloud type in that region, and then from the top two most common, etc. This is
shown in Figure 4. In panel (b), we see that in the northwestern corner of our region of interest, the top two cloud types (in
       this case, suppressed and clustered Cu) account for more than 90% of all scenes. This would suggest that any further
       differentiation into more specific cloud subtypes would be most effective if focused on this region. Panel (c) and (d) show that
       the region with the greatest variability in cloud type is the zonal band near 45°S, as well as the subtropical Sc-Cu transition
       region near 15 °S.



## 3.2 Sample case

To better illustrate the scale on which the classifications as well as the underlying data exist, Figure 5 shows a case study from July 22nd, 2015. Each panel shows the classifications in colored circles, marking the center of each rectangle of MODIS image on which the classifications are carried out (see Yuan et al., 2020, for additional details on classification).

The scene was selected to highlight suppressed and clustered types. In panel (a), a roughly 200 km by 400 km region of enhanced cloud in the lower middle of the scene is identified as clustered Cu, surrounded by suppressed Cu scenes. A misidentification of sun glint as solid stratus is evident as well (though Figure 3 shows that very few tropical scenes, in which sun glint is found, are identified as solid stratus, suggesting that there is no significant impact on the climatology of classifications). Panels (b) and (c) show the surface divergence as observed from ASCAT and from the MERRA-2 reanalysis; as the ASCAT overpass time at 9:30 is 4 hours ahead of the MODIS Aqua observation time, there is a slight geographic mismatch. Importantly, both surface divergence plots show strong convergence (in blue) in the clustered region, and surrounding divergence. Note also the noisy nature of the ASCAT observations, as well as the narrow swath of ASCAT not allowing matches with many (approximately half) of the classifications. Panel (f) shows the large-scale divergence as inferred from the 700 hPa vertical motion. Although there is some convergence aloft at the southern boundary of the scene (where the MERRA-2 surface convergence is strongest), the rest of the clustered region shows slightly enhanced subsidence aloft, in contrast to the surface conditions, which we will see later is also the mean behavior for clustered scenes. MODIS gives cloud top pressures between 800 and 700 hPa (not shown) at around 15˚N, 138˚W (where the divergence is strongest), consistent with the schematic model in BB17 (their Figure 10). This divergence may potentially represent the outflow from the aggregated convection in this clustered region.

Panels (d) and (e) show the AMSR-2 precipitation and moisture retrievals, respectively. The clustered (suppressed) classifications are consistently associated with a moist (dry) CWV anomaly, and precipitation is only found in the clustered regions. Overall, the mesoscale anomalies in the presented datasets are clearly resolved on the spatial scales of the classifications. There are also classification edge cases where a human observer would struggle to clearly identify a scene as suppressed or clustered, however in aggregate the machine learning classifications are consistent with what a human would label; performance evaluation is presented in Yuan et al. (2020).

## 3.3 Classification radiative properties

As the climatological relevance of marine low clouds is based in large part on their radiative effect, it is worth knowing the variability in radiative properties between the different categories. Figure 6 shows the low cloud fraction of each cloud type, with closed MCC being the highest and suppressed MCC the lowest. The mean cloud fraction across all scenes (black dot at right of panel a) also shows that the Cu-vs-Sc cloud types also split tidily into the below-average and above-average cloudy scenes, for this particular sample. The mesoscale cloud fraction anomaly (represented by the difference between the small diamonds and circles, for each type) show that, on average, the scenes we classify are slightly cloudier than their



surroundings. This is most pronounced for the closed MCC, and most likely a result of the filtering of scenes with very low cloud. The only exception is suppressed Cu, which is associated with a low CF anomaly. The same is true when comparing to the climatological cloud fraction (small squares); there is a high bias in the cloud fraction, again most likely due to the fact

that we can only classify cloudy scenes.

Panels (b-d) show the composite net CRE of the various cloud types. In panel (b) the overall frequency of each cloud type in our dataset is broken down by year (2014-2016). Together, clustered and suppressed Cu scenes account for more than half of all scenes. Panel (c) shows the CERES net CRE as calculated in section 2b) for each type and year, as well as the mesoscale and climatological value; the CRE broadly mirrors the cloud fraction (for marine low clouds, the radiative effect is

almost entirely a shortwave effect). The total cooling averaged over all scenes is shown as the black dots in panel (c); on average, a scene in our dataset has a net CRE of ~-113 W m$^{-2}$. Note that due to the specific sampling strategy (only considering scenes with low cloud, without too much overlying high cloud), and the fact that we composite instantaneous daytime values that are not weighted by the global frequency of occurrence of our cloud types, our CRE for marine low clouds is approximately an order of magnitude larger than the global value found by L'Ecuyer et al (2019).

The above difference between instantaneous local and global values underscores the fact that when considering the radiative importance of different cloud types, both frequency and mean CRE at the time of occurrence are relevant. Specifically, when considering the Cu-cloud types (clustered and suppressed), while both have a relatively low mean instantaneous CRE for any particular scene (particularly the suppressed scenes), these two types are also the most frequently-occurring in our dataset, due to their dominance in the tropics and subtropics. Therefore, calculating the frequency-weighted

CRE (panel d), which is simply the product for each year of the data in panel (b) and panel (c), is appropriate. This represents the fraction of total cooling, over all scenes, by a particular cloud type. Thus open-MCC, despite having a mean net CRE of -100 W m$^{-2}$, only accounts for ~5 W m$^{-2}$ of the total cooling of all scenes in our dataset (approximately 4%); while these scenes have high CFs and therefore net CRE, they are infrequent, more so in this classification compared to previous work. For the clustered and suppressed types, the importance of understanding their drivers is highlighted in panel (d); clustered Cu scenes

have five times higher contribution to the net CRE than suppressed Cu scenes.

**3.4 Composite analysis**

Figure 7 and Figure 8 are similar to Figure 6, showing composites of meteorological variables by cloud type, as well as synoptic and climatological averages (where seasonal mean values for a given location are composited instead of instantaneous values). For both these figures, we can estimate the variability between types explained by differences in

geography by comparing the mesoscale averages (circles) to the climatological averages (squares). For instance, for every cloud type, there is almost no bias between the mesoscale and climatological averages of sea surface temperature (SST, panel a). In other words, variation in SST between scenes is almost entirely explainable by the variation in geography. The suppressed scenes occur over the warmest waters, and the closed MCC over the coldest. The same is largely true for EIS, which is determined in part by SST. This is not surprising given the geographic distributions of the cloud types seen earlier and





climatological gradients in SST and EIS. What this tells us, however, is that there is no strong evidence for sub-seasonal time scale perturbations to SST or EIS coinciding with variations in cloud type. We can also compare the mesoscale averages to the 10˚ synoptic averages to assess whether any mesoscale anomalies are coincident with cloud type variability. However, an important caveat to bear in mind is the bias introduced by our sampling strategy: only scenes with some low cloud and not too much high cloud are considered, whereas the surrounding scenes are not similarly constrained. These biases are best identified

from the black 'all scenes' markers. For instance, we notice in panel (d) that averaged over all scenes, RH700 is biased low by 3%, most likely due to preferential selection of scenes with little high cloud (and therefore the free troposphere is biased dry). This bias is also applicable to the climatological comparison. The dry free troposphere (FT) anomaly relative to the synoptic (and climatological) averages in e.g. the closed MCC scenes can be explained by this sampling bias and is not indicative of some mechanism in a drier FT yielding closed MCC clouds.

With that caveat in mind, Figure 7 shows that closed-MCC and to a lesser extent disorganized MCC is associated with a significant mesoscale anomaly in EIS (consistent with Muhlbauer et al., 2014). Solid stratus is associated with a positive anomaly in vertical motion and RH700 relative to climatology, but not a mesoscale one, indicating that this link is driven by synoptic features; manual inspection shows that many scenes identified as stratus are indeed associated with frontal systems. Both closed and open MCC are associated with strong subseasonal anomalies of enhanced subsidence, though again the

absence of an anomaly relative to the synoptic mean indicates that these are larger features, likely associated with variability in the position of the subtropical high.

     Aside from the mesoscale and subseasonal anomaly analysis, a key result is that clustered and suppressed types are poorly separated by the variables in Figure 7; they have virtually identical EIS distributions, and though suppressed scenes are associated with slightly higher SST, large-scale divergence, and lower FT humidity, there is not much separation between them in this phase space, especially relative to the variability between all cloud types, and these small differences are consistent

with their slightly different geographic distributions. In contrast, EIS is an excellent discriminator between the stratiform MCC types.

     Composite analysis of the surface divergence, however, is much more helpful at distinguishing between the Cu cloud types. This is evident from Figure 8, panels (a) and (b). From the ASCAT composite data, the strongest surface divergence is

associated with suppressed scenes, and the strongest convergence with the clustered scenes. When using MERRA2 data, the only difference is that the closed MCC cases have slightly stronger divergence, yet the clear separation between Cu types remains. Additionally, the surface divergence signal is clearly a mesoscale one and not explained by climatological differences, particularly for the convergence associated with clustered scenes; the synoptic environment shows broad divergence.

     Having calculated both the 700 hPa large-scale and surface divergence, we can subtract the former from the latter to

estimate a boundary-layer anomaly divergence. If near-surface divergence purely reflects the large-scale subsiding flow, with no additional low-level circulation, we would expect this anomaly to be small. Figure 9(a) shows this surface level anomaly, using both the MERRA2 and ASCAT winds. The large positive anomaly for suppressed Cu scenes indicates that the bulk of the divergence as a result of near-surface circulations rather than those extending over a deep layer of the lower troposphere;



similarly for clustered Cu, the surface convergence together with mean large-scale divergence indicate a shallow circulation,
as seen in the case study of Figure 5.

Considering AMSR-2 retrievals, the rain rate shows a very clear separation between clustered and suppressed cloud types, with a strong positive (negative) mesoscale anomaly for clustered (suppressed) Cu of around 0.4 mm day$^{-1}$. Similar qualitative results are found for conditional rain rates and rain probabilities (not shown). It is worth noting that the resolution of the precipitation data is approximately 4 km, so the smallest clouds will not be resolved. The column water vapor results
are interesting as well; consistent with the warm SSTs, both Cu cloud types occur in areas of high column water vapor. The mesoscale anomalies, however, are consistent with the BB17 presentation: clustered scenes are slightly moister than their environment and suppressed scenes slightly drier. This is difficult to identify in Figure 8(d), so Figure 9(b) shows just the mesoscale anomaly for all cloud types and makes clear that the suppressed scenes are the most anomalously dry and the clustered scenes most anomalously moist. Although the moisture anomalies of the LES in BB17 were larger than those found
here, this may be due to their mean state being moister. One finding from that work is that amplitude of aggregation-associated moisture anomalies tended to scale with the mean state CWV, and so we expect the higher mean state moisture in BB17 would occur with larger moisture anomalies.

### 3.5 Aircraft observations

Figure 10 shows the depth of the boundary layer and degree of decoupling (using the $\alpha_q$ metric from Wood and
Bretherton, 2004) based on CSET aircraft profiles. For each aircraft profile, the cloud type classification which covers that profile is selected for compositing, and so the profile represents a random estimate of depth or decoupling within that scene. Here the sample sizes are much smaller than the composites of satellite and reanalysis data, and so the full histograms are shown (smoothed using kernel density estimation) to highlight the uncertainty. Adopting a Lagrangian perspective which accounts for the boundary layer evolving down the trade winds through the Sc-Cu transition, there is boundary layer deepening
and decoupling from stratus through closed, disorganized, and open MCC; in particular the degree of decoupling between closed and open MCC is very pronounced, with the former being the most coupled and the latter the most decoupled. However, this evolution breaks down for the Cu-type boundary layers, which are neither deeper nor more decoupled than open MCC. This is not surprising, as the inversion at the top of the surface mixed layer where Cu clouds form will persist as the decoupled Sc layer is eroded, such that the remaining boundary layer stays shallower and well-coupled to the surface. Also important to
note is that, as with EIS and SST, clustered and suppressed types are difficult to distinguish by their depth and decoupling state, though clustered scenes are marginally deeper in panel (a).

### 4 Conclusions

In this study we have analyzed the characteristics of the marine boundary layer for six different cloud types, based on a novel machine-learning based cloud classification dataset. Specifically, we assessed whether the observations of clustered




and suppressed cumulus are consistent with previous modeling of mesoscale aggregation of shallow cumulus. The key findings are as follows:

- The six cloud types represent distinct MBL regimes, based on their geography and environmental conditions.
- The anomalies in cloudiness, column water vapor, circulation, and precipitation are consistent with the BB17 LES results and conceptual model for mesoscale shallow aggregation.
- Suppressed and clustered Cu scenes are most clearly separable by looking at surface wind divergence, and this signal is apparent in both satellite retrievals as well as in the MERRA-2 reanalysis.

This last point touches on a more general conclusion, namely that, at least for the variables considered, mesoscale anomalies in meteorological variables are more pronounced for the cumulus types than the stratiform MCC types; this is true for CWV, precipitation, and surface divergence. For discriminating between the MCC types, EIS, depth and decoupling are
more useful; in stratocumulus regions, these variables have been shown to correlate strongly with each other and with cloud cover (Wood and Bretherton, 2004; Wood and Hartmann, 2006).

Though it is tempting to conclude that surface divergence is such a good discriminator because the mesoscale aggregation described in BB17 is likely the most important determinant of cloud variability, we must also bear in mind that, along with precipitation, it is more an 'internal' boundary layer predictor than most of the other predictors, e.g. EIS or SST, and therefore
better coupled to other MBL state variables (e.g. cloud fraction). Additionally, it is also much more directly observed and resolved at finer scale than e.g. 700 hPa vertical motion, and therefore has a lower observational uncertainty. That being said, the strong consistency between the observations and the BB17 LES modeling of mesoscale shallow convection does indicate that this process is an important driver of cumulus-dominated MBL cloud variability.

There are a few limitations on the generalizability of these results. The first is that we have only considered the SEP and
NEP regions, and other clouds, particularly those in the warmer Trade wind regions of the western ocean basins, may have different MBL characteristics. The second is that we have only considered daytime behavior and cannot account for diurnal variability in cloud type. The observations from aircraft data where limited and did not extend south of Hawaii or north of California. Lastly, we have not examined in depth the role of SST in determining cloud type. This is not because it is unimportant (on the contrary, it is a key driver of many MCC variability; see McCoy et al., 2017), but rather because it does
not vary much at mesoscale and short time scales.

With regards to climate modelling, CRE for different cloud types largely mirrors cloud fraction. While the CRE between suppressed and clustered types is very different, it remains to be seen whether the process of shallow convective aggregation affects synoptic-scale mean cloud cover or CRE. Given that models capable of reproducing such shallow aggregation are now able to run at global scales (Bretherton and Khairoutdinov, 2015), this question is best answered using simulation studies.



**Acknowledgements**

We gratefully acknowledge our colleagues at University of Washington for feedback and helpful discussion. Funding for this research was provided in part by the NASA MEaSUREs program (award number 80NSSC18M0084).

**Data Availability**

MODIS reflectance data for this work is available at https://modis.gsfc.nasa.gov/data/dataprod/mod02.php. ASCAT data
is available from http://www.remss.com/missions/ascat/. AMSR-2 water vapor data is available at http://www.remss.com/missions/amsr/. CERES SYN1deg data is available at https://ceres.larc.nasa.gov/data/. MERRA-2 data is available at https://gmao.gsfc.nasa.gov/reanalysis/MERRA-2/data_access/. CSET aircraft data is available at https://www.eol.ucar.edu/field_projects/cset.

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





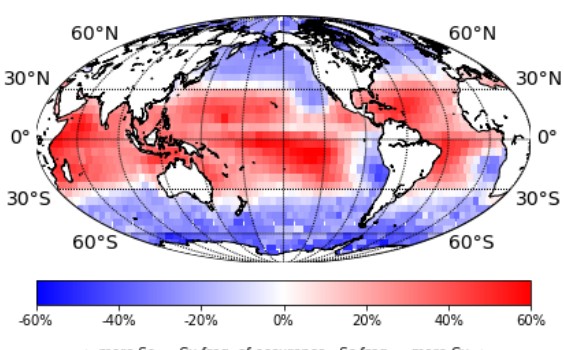

**Figure 1: Difference in relative frequency of occurrence of cumulus and stratocumulus cloud types per Hahn et al. (2001) definitions from ship-based observations. Red areas highlight Cu-dominated MBLs, while blue regions have more Sc cloud.**

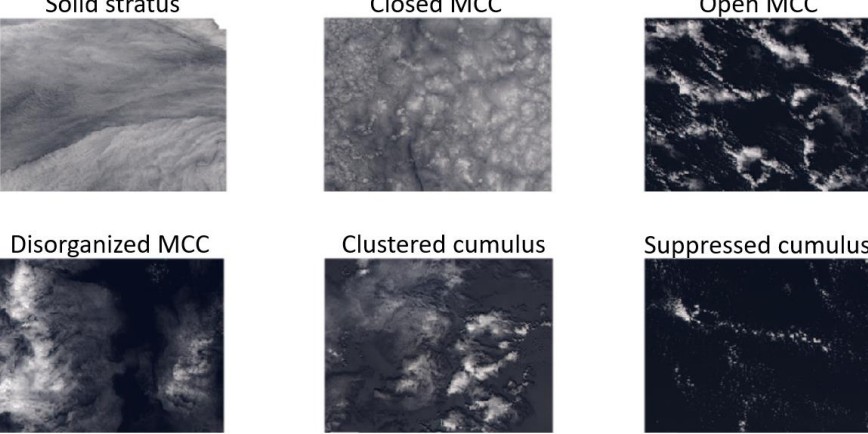

**Figure 2: Typical examples of scenes belonging to each of our classification categories.**




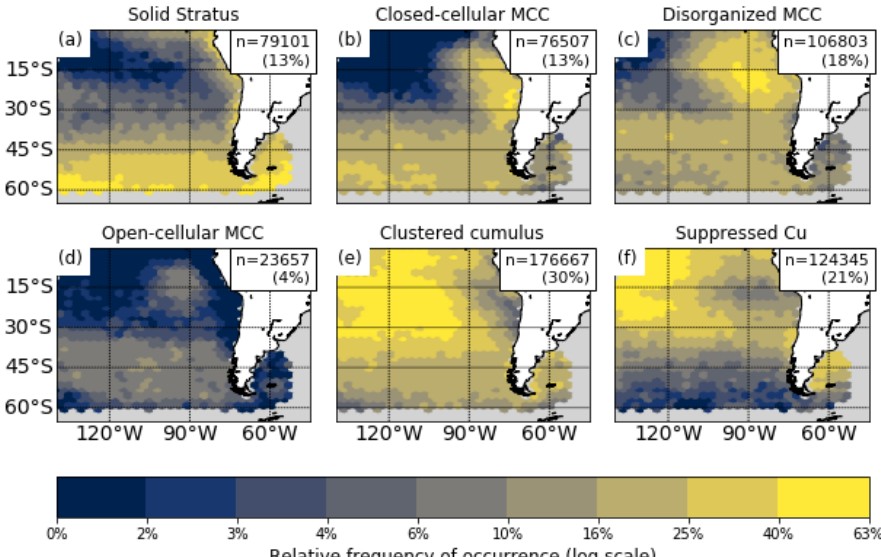

**Figure 3: Relative frequency of occurrence of each cloud type on a logarithmic scale. In the upper right corner of each panel, the total number of classifications over three years (2014-2016), as well as the total fraction of scenes of each type, is shown. Grey areas are where fewer than 200 scenes are sampled.**


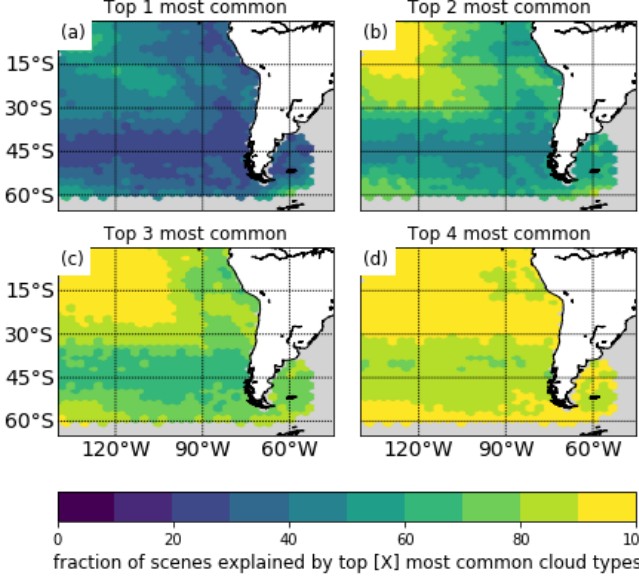

**Figure 4: The fraction of cloud scenes for each grid point which are represented by (a) the most common cloud type for that grid point, (b-d) the top 2 through 4 most common cloud types. Grey areas are where fewer than 200 scenes are sampled.**


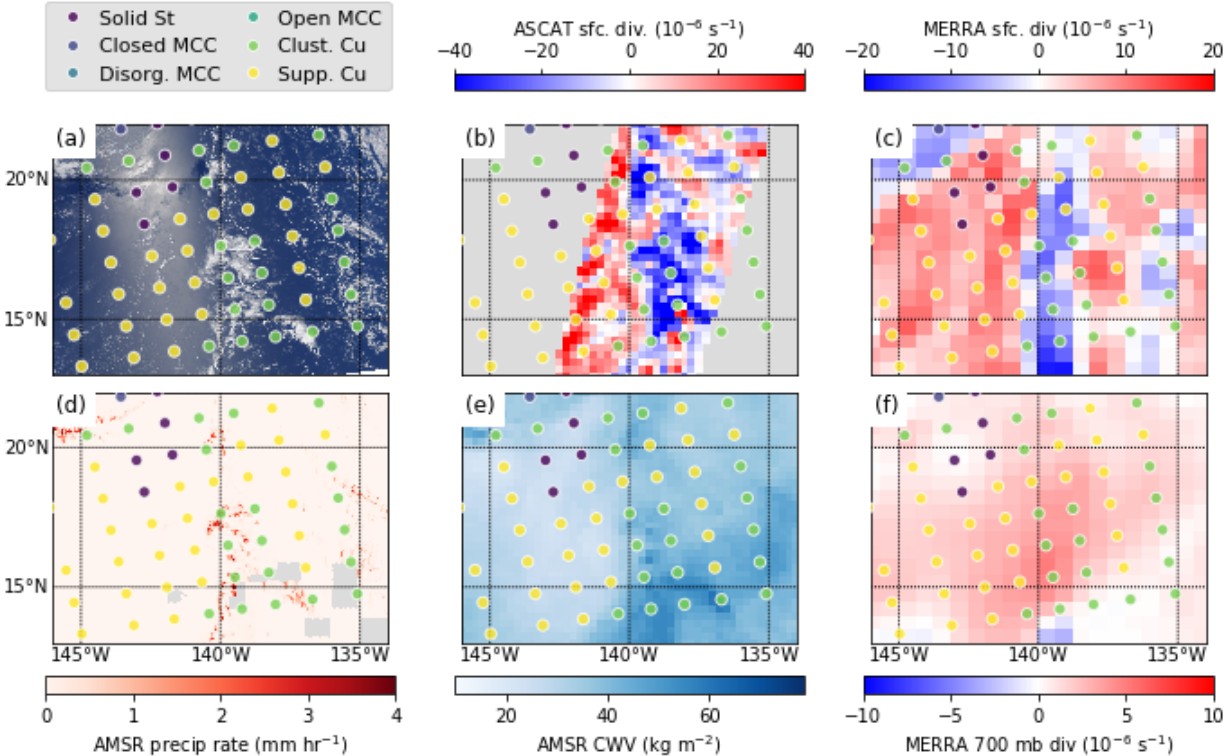

Figure 5: Sample region observed on July 22, 2015 showing classifications (every panel, in colored circles). (a): MODIS true-color reflectance; (b) ASCAT surface divergence; (c) MERRA-2 surface divergence; (d) AMSR2 89 Ghz precipitation rate; (e) AMSR2 columnar water vapor; (f) MERRA-2 700 hPa divergence.





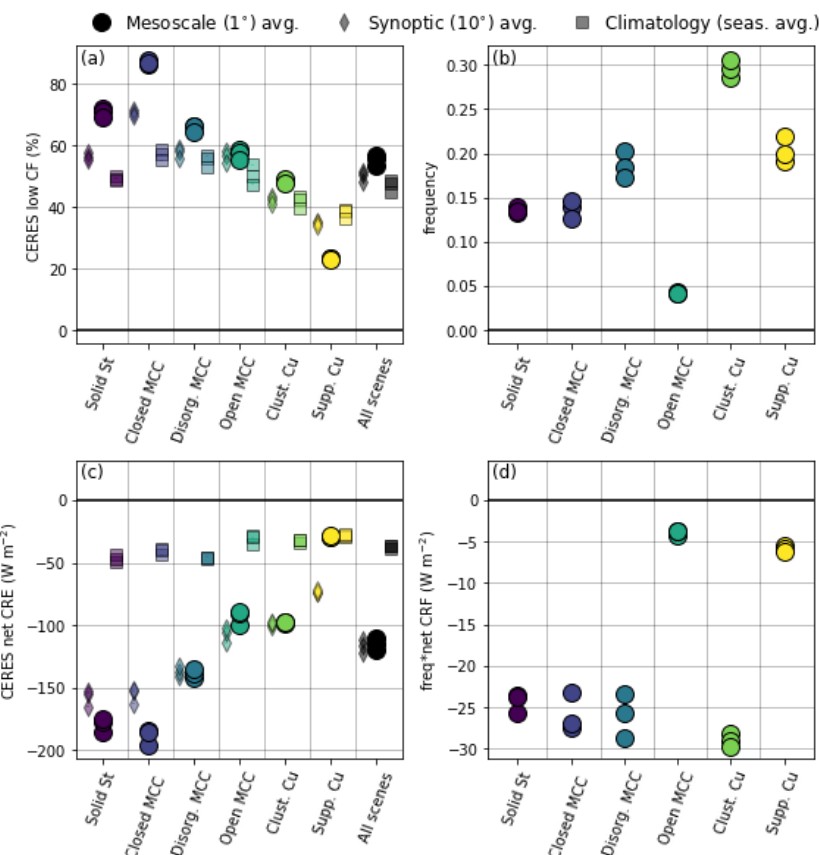

**Figure 6: Cloud radiative properties by cloud type: a) CERES cloud fraction; b) cloud frequency of occurrence, c) average net CRE per cloud type, from CERES; d) frequency-weighted net CRE. Each set of three markers is for the 3 years (2014-2016) used. For panels (a) and (c), the mesoscale, synoptic, and climatological averages are shown using circular, diamond, and square markers respectively (see Section 2e).**




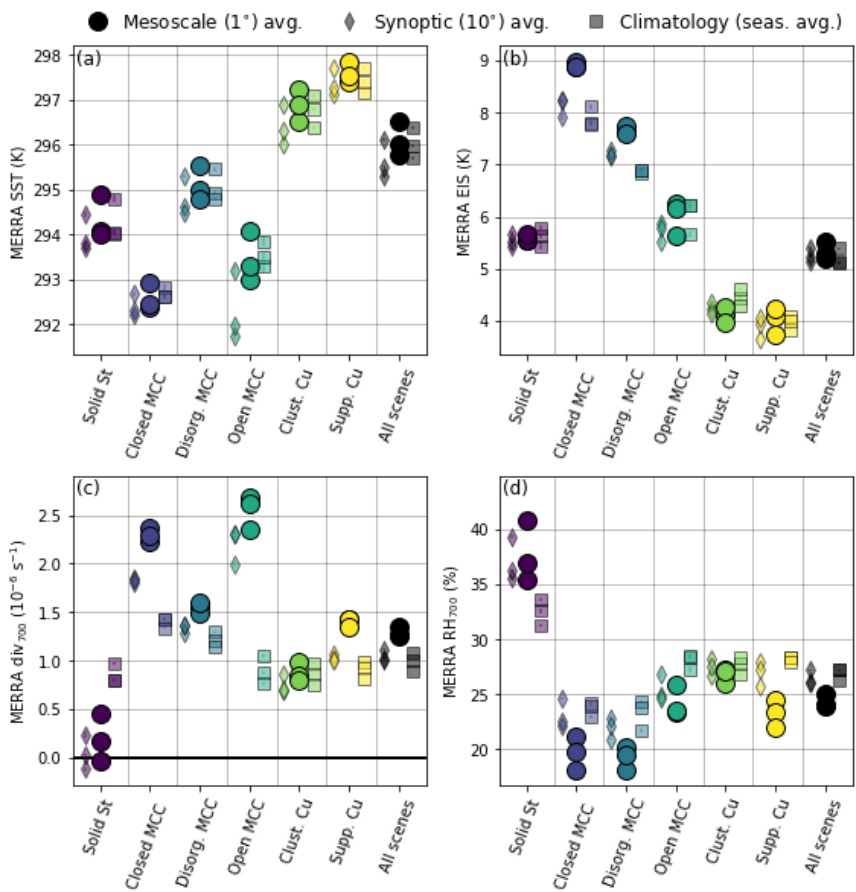

**Figure 7:** Same as Figure 6a, but for (a) MERRA2 sea surface temperature; (b) MERRA2 estimated inversion strength (EIS); (c) MERRA2 700 hPa divergence; (d) MERRA2 700 hPa relative humidity.



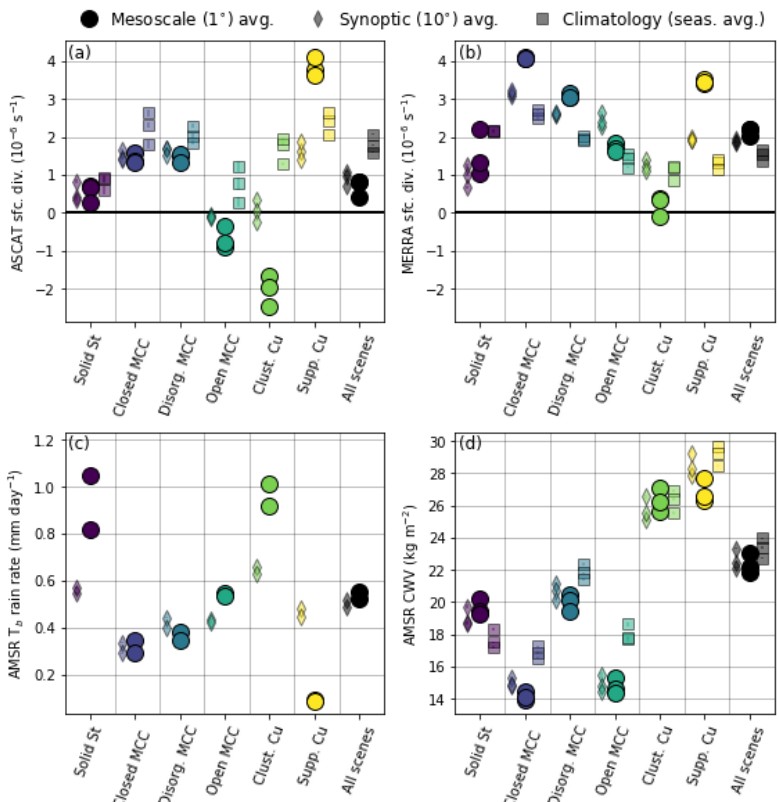

**Figure 8: Same as Figure 7, but with (a) ASCAT surface wind divergence; (b) MERRA2 surface wind divergence; (c) AMSR rain rate; (D) AMSR column water vapor.**

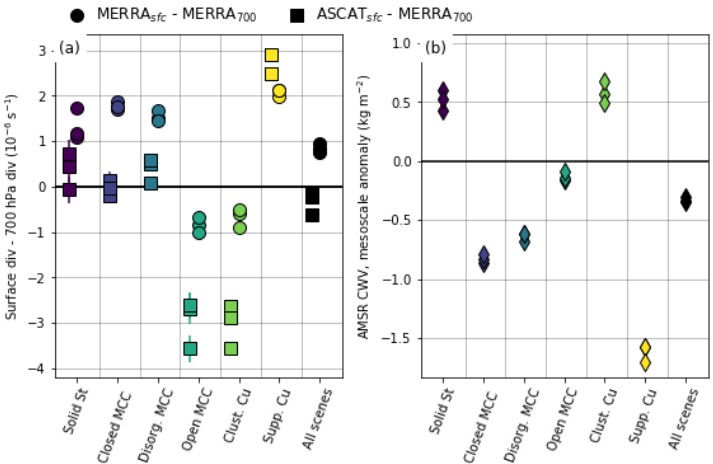

**Figure 9: (a) Surface divergence anomaly from 700 hPa; circles are based on MERRA2 surface winds, squares are based on ASCAT surface winds. (b) AMSR column water vapor mesoscale anomaly.**





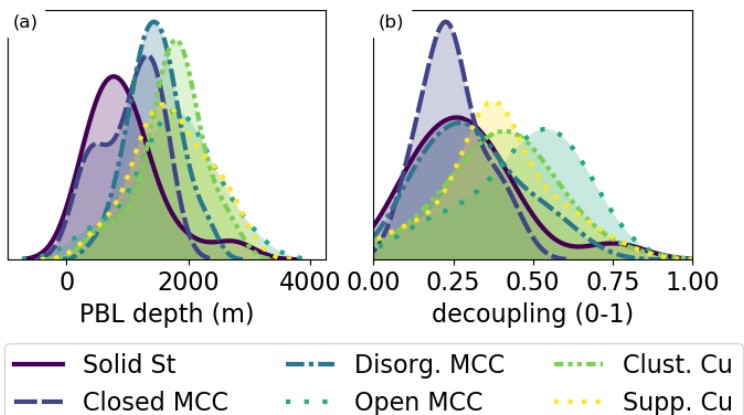

**Figure 10: Histograms of (a) boundary layer depth and (b) boundary layer decoupling index from CSET flight and dropsonde observations.**
