# Peer review of "Identifying meteorological influences on marine low cloud mesoscale morphology using satellite classifications"

_Atmospheric Chemistry and Physics, 2020_

## Referee Comment (RC1) · Anonymous Referee #1 · 27 Dec 2020

This paper analyzed the marine low cloud mesoscale morphology in the southeastern Pacific Ocean using a machine-learning generated classification dataset that have been published in another paper. This paper shows that the different cloud types represent the distinct MBL regimes and finds that the cloud meteorological properties are consistent with prior knowledge. However, the title is very confusing due to the "using deep learning classifications". As a reviewer or reader in the field of machine learning, it's easy to mislead what this paper focus on is deep learning classifications. Nevertheless, they only use the dataset by deep learning classification (published in another paper) for further analysis. Therefore, I would recommend for a major revision.

[Figure]

1. The title of this paper is very confusing for me because it claims using deep learning methods to identify the meteorological influences on marine low cloud mesoscale morphology. It is easy to mislead what this paper focus on is "deep learning". However, this paper just analyzes the results of deep learning achieved by another paper. Although the further analysis can help deeply understand the insight behind the deep learning and overcome the black-box issue, the details about how the deep learning method conduct the classification are described in another paper. In this paper, the authors motioned we can see the paper of Yuan et al., 2020 in P3L89 and P7L193. I think it is not friendly for readers and it could be better if the authors could briefly describe the machine learning method in this paper.

2. P3L86: The cloud types are classified as stratus cloud, closed MCC, open MCC, disorganized MCC, clustered cumulus, and suppressed cumulus. Could you explain why the clouds are divided into these six types?

3. P4L118, please provide the reference of CERES.

4. P4L125, please provide the reference of MERRA2.

5. P5L150: The appearance of Figure 6 and Figure 7 may not be very appropriate because the appearance of figures would be better in order. Here figure 6 and figure 7 is ahead of figure 4 and figure 5. It could be better if the authors adjust the order of figures.

6. P6L170: Before discussing the characteristics of the cloud types, it could be better if briefly describing the accuracy of the deep learning results. Because the credibility is the baseline for further analysis of the deep learning dataset.

7. P6L185: For the figure 4, panel a) is not mentioned in texts.

---

## Referee Comment (RC2) · Anonymous Referee #2 · 6 Jan 2021

Summary Satellite images of fields with low clouds are categorized with a machine learning technique to identify fields with solid stratus, closed-cellular mesoscale cellular convection (MCC), disorganized MCC, open-cellular MCC, clustered cumulus or suppressed cumulus.This enables an assessment of their frequency of occurence and their impact on radiation is quantified. It is explored whether conditions like the large-scale divergence of the horizontal wind, bulk thermal stability or humidity may be factors controlling the prevailing cloud regime. This approach demonstrates that 1) stratus clouds are more likely to be associated with frontal systems rather than synoptic high pressure systems with large-scale subsidence whereas 2) the bulk thermal stratification (as quantified by the estimated inversion strength EIS) nicely sorts mesoscale

cellular convection from suppressed cumulus.

The findings are key to anyone with an interest boundary-layer clouds, either from an obserational or a modeling perspective. The paper is well written, to-the-point, and the figures are clear. The strength of the present work is the use of a large amount of satellite images, including satellite observations of radiative fluxes, in addition to aircraft observations to analyse the vertical boundary-layer structure as well as reanalysis data to assess the large-scale subsidence and divergence.

I believe the manuscript is almost ready for acceptance in its present form. However, I have some questions and remarks that the authors may find useful for improving the manuscript.

Remarks ————

l101: 'surface wind divergence is derived...' . Perhaps the equation for the conservation of mass can be presented to illustrate the relation between the large-scale divergence of the horizontal winds and the large-scale vertical motion (subsidence), including the sentence from l133 "The terms large-scale divergence and subsidence are used interchangeably throughout;"

l202: "Panel (f) shows the large-scale divergence as inferred from the 700 hPa vertical motion." The large-scale divergence Div =partial u/partial x + partial v/partial y, so why is Div not diagnosed from the local horizontal velocities at 700 hPa?

l218: 'suppressed MCC' This seems like an error and should be replaced by suppressed cumulus?

l218-220: 'The mean cloud fraction across all scenes (black dot at right of panel a) also shows that the Cu-vs-Sc cloud types also split tidily into the below-average and above-average cloudy scenes,' Is it meant that cumulus have low cloud fraction as opposed to the mesoscale mean values (the black dots?) for MCC and stratus?

l234: paper of L'Ecuyer et al (2019) is missing in the bibliography

3.5 Aircraft observations and decoupling parameter alpha_q 1) It may helpful to the reader not familiar with alpha_q to give a short explanation of this quantity. 2) alpha_q measures the difference between cloud-layer and subcloud layer properties. How is this computed for fields of shallow cumulus clouds where moisture and potential temperature vary with height in the cloud layer? 3) I consider Fig. 10 which shows PDFs of the PBL depth and the decoupling parameter a highlight of the present work. I was wondering whether their joint PDF might display some additional useful information? This question is motivated by two recent intercomparison papers that diagnose alpha_q and cloud layer depth for both LES and single-column model results of stratocumulus-to-cumulus transitions (Fig 6 in De Roode et al. 2016 and Fig. 14a Neggers et al. 2017, respectively). It is found that the LES results deviate somewhat from the fits as proposed in Wood and Bretherton (2004). A joint PDF may shed some light on the question whether the modeling results fall within the observations as presented in the manuscript?

Cumulus versus stratocumulus Lock (2009, factors influencing cloud area at the capping inversion for shallow cumulus clouds) finds from LES results that a parameter called kappa, and which depends on the inversion jumps of potential temperature and the total specific humidity, controls cloud cover. Would it be possible to verify the relation he found from the satellite images and the aircraft observations?

Fig. 2: - Could the horizontal sizes of the images be mentioned in the caption?

- Disorganized MCC and clustered cumulus bear some similar structures. Could the authors briefly describe which main criterion identifies these two regimes?

Fig. 5: - although not present in cloud scene, the colors of disorg MCC and Open MCC as shown in the legend appear almost the same on my screen

- the range of values shown for the divergences in figs b and c are different. wouldn't it be neater to use the same axis ranges as they both display the same quantity?

[Figure]

l342 : where -> were

---

## Author Comment (AC1) · 11 Mar 2021

Response to reviewer comments, "Identifying meteorological influences on marine low cloud mesoscale morphology using deep learning classifications", Mohrmann et al.

Response to reviewer #1:

We thank the reviewer for their time and helpful comments on the manuscript. Based on their feedback, we have the following response and revisions to the manuscript:

**Comment (1), also (6):** The reviewer raises a very good point regarding the confusing wording of the manuscript title, and in hindsight we understand why a reader could be misled in thinking that deep learning was used to identify meteorological influences, and not merely for creating the classification dataset used. A title change is warranted, and we have altered the new title to be "Identifying meteorological influences on marine low cloud mesoscale morphology using satellite classifications". As the reviewer points out, the description of the classification dataset is primarily carried out in Yuan et al. (2020), and so no emphasis is needed in the title of this manuscript. This manuscript focuses primarily on the meteorology, so we did not describe extensively the methods behind the classification or its accuracy. Still, we agree with the reviewer's comment (both comments (1) and (6)) that the accuracy of the deep learning and a brief description of its methodology and accuracy is warranted. The following language has been added to section 2.1, paragraph 1, in the description of the classification dataset:

Old text: [list of scenes] …These scenes are then used to train a convolutional neural net, which in turn is run near-globally on such MODIS oceanic scenes., and a detailed description of the classification dataset and training can be found in Yuan et al. (2020).

New text: [list of scenes]. These categories were chosen by examining the morphological climatologies in Muhlbauer et al. (2014) and studying regions where there was little variability in morphology category (primarily the tropics, where disorganized MCC dominated), and identifying additional commonly occurring cloud morphologies. These (clustered and suppressed Cu) were then added to the pre-existing cloud categories, along with a homogeneous stratiform category initially used in Wood and Hartmann (2006). Examples of these types can be found in **Error! Reference source not found.**.

The scenes were then used to train a convolutional neural net (CNN) using as input the image of scene visible reflectance. A full description of the machine learning training and model evaluation can be found in Yuan et al. (2020). These authors found that average model precision evaluated on a test set was approximately 93% across all categories. Open-MCC had the lowest precision, most likely because it was the lowest-frequency category. The largest source of model confusion was between disorganized MCC and clustered Cu, which is unsurprising given the similar appearance of these categories.

Comment (2): Please see the first paragraph added above, where we have added additional explanation about the origins of these cloud categories. Additional information about them can be found in Yuan et al. (2020) and we do not see the value of repeating too much of that discussion in this work.

Comment (3): We have added the appropriate citation for CERES (Doelling et al., 2013)

Comment (4): We have added the appropriate citation for CERES (Gelaro et al., 2017)

Comment (5): While we understand the reviewer's point that Figures 6 and 7 are mentioned ahead of Figures 4 and 5, their mention in section 2.5 only serves as a possible reference for the methods described and not discussed in any depth. The analysis that corresponds to Figure 6 and 7 indeed comes

later, in section 3.3, following the analysis of Figure 4 and 5 (in Sections 3.1 and 3.2 respectively). We feel that ordering the figures this way makes for a more coherent reading of the paper; the alternatives would be to move the methods in section 2.5. to 3.3 (breaking up the methods), or moving Figures 6 and 7 up, resulting in much backtracking for the reader when they reach section 3.3.

Comment (6): Addressed in response to comment (1).

Comment (7): The following line has been added to section 3.1: "Panel (a) shows the fraction of scenes covered by the dominant cloud type for that grid box" to address this omission, thank you.

---

## Author Comment (AC2) · 11 Mar 2021

Response to reviewer comments, "Identifying meteorological influences on marine low cloud mesoscale morphology using deep learning classifications", Mohrmann et al.

Response to reviewer #2:

We thank the reviewer for their time and helpful comments on the manuscript. Based on their feedback, we have the following response and revisions to the manuscript:

Comment on l101, l133, l202: By mass continuity, the horizontal divergence at 700 hPa would only give us the *local* vertical divergence dw/dz (or dω/dp in pressure coordinates) at 700 hPa, and while we do not expect very strong gradients of large-scale divergence with height in the marine lower troposphere, this value will be somewhat sensitive to the level chosen. The quantity $w_{700}/z_{700}$ results from the integration of dw/dz from the surface to 700mb, and so represents the mean horizontal divergence over that layer, making it more suitable as an estimate of large-scale divergence. To make this point clearer, we have amended this paragraph as follows:

"Note that this large-scale divergence is not the horizontal divergence at 700 hPa, but rather the mean divergence from the surface to the 700 hPa level; this follows from the mass continuity equation by considering a column of air from the surface (where vertical motion is 0) to 700 hPa. The terms large-scale divergence and 700 hPa subsidence are used interchangeably throughout; divergence is plotted instead of subsidence to allow for more straightforward comparison with surface divergence. As surface pressure varies with time, the second equality is only approximate."

Comment on l218: Corrected typo, thank you.

Comment on l218-220: The reviewer correctly interpreted this sentence and is perhaps wondering if there was anything more to this point; we have added a parenthetical "(as expected)" to this sentence to indicate there is no deeper point being made.

Comment on l234: Corrected by adding reference.

Comment on S3.5: regarding alpha_q, we have added the following text: The parameter $\alpha_q$ is a measure of relative resemblance of upper boundary layer moisture to the lower FT and lower boundary layer, with a value of 0 indicating a perfectly well-mixed boundary layer and a value of 1 indicating a perfectly decoupled boundary layer where the upper BL moisture is equal to the lower FT moisture.

$$\alpha_{qT} = \frac{q_T(upper\ BL) - q_T(lower\ BL)}{q_T(lower\ FT) - q_T(lower\ BL)}$$

For a given profile, the thermal inversion height is estimated using the maximum lapse rate, with the inversion being the layer where the lapse rate deviation from a moist adiabat exceeds 25% of maximum deviation (this was tuned to agree with a visual assessment of the inversion layer and worked well for all profiles). Upper and lower BL in the $q_T$ equation are taken as the top and bottom 25% of the BL depth, while the lower FT starts 500m above the inversion top.Regarding question 2 on this section, this is clarified in the text added above; the inversion used for diagnosing MBL depth is always the strongest thermal inversion, which tended to occur at the top of the remnant cloud layer in trade-like shallow Cu profiles. The result is that the boundary layers appeared highly decoupled as this layer has

been subject to ample dry entrainment during the transition. There were some cases where the upper layer was thoroughly eroded by dry entrainment making the diagnosed inversion was much shallower.

Regarding question 3 on this section, we reprocessed the profiles to include a surface mixed layer depth using the (near-)surface-derived LCL, consistent with Wood and Bretherton (2004). This allows for a more apples-to-apples comparison with the figures referenced by the reviewer showing model results of the same quantities. We include the fit from Park et al, 2004 shown in all 3 previous papers (though we note that Neggers et al. (2017) used a lower gamma value). It certainly seems that our profiles are consistent with the results in Wood and Bretherton, though

we do not have enough sample size to say anything more insightful regarding the slight disagreements between the LES results and the plotted fit in de Roode et al. 2016.

Regarding the comparison to Lock (2009), we did briefly attempt to validate those results, but could not find a strong correlation between kappa and low cloud fraction. The most likely explanation for this is that in our observations, both kappa and cloud fraction are measured roughly simultaneously, whereas the simulations in Lock allow for cloud adjustment. It is also possible that kappa is sensitive to noise resulting from messy real-world profiles, and so we cannot say anything one way or another about its role in controlling cloud fraction.

Comment on Fig 2: added "Image scale is roughly 100 km across." to caption.

For the discussion of clustered vs disorganized MCC, the following text has been added to section 2.1: "The primary difference between these two types is that disorganized MCC represents a regime with cellular convection at some characteristic scale, though not organized clearly into open- or closed-cell regimes, while clustered Cu represents aggregated convection at a variety of scales within a scene. When distinguishing between these two types during manual labelling, scene large-scale context proved helpful."

Comment on Fig 5: This may be a screen issue, though we do agree that for Figure 5, the colors are not as easy to distinguish, though as the reviewer notes it is a moot point as no disorganized MCC or clustered Cu occurs in this scene. The colors are selected for consistency with the rest of the plots, where they do not present an issue, and so we will keep them as is.

Regarding the differing color scales between the two divergence plots, we have updated the plots so that both ASCAT and MERRA surface divergence are on the same color scale.

Comment on l342: typo corrected, thank you.

---

## Author Response (AR2)

Response to reviewer comments, "Identifying meteorological influences on marine low cloud mesoscale morphology using deep learning classifications", Mohrmann et al.

We thank both reviewers for their time and helpful comments on the manuscript. Based on their feedback, we have the following revisions to be manuscript:

Response to reviewer #1:

**Comment (1), also (6):** The reviewer raises a very good point regarding the confusing wording of the manuscript title, and in hindsight it is very understandable that a reader could be misled by it to think that deep learning was used in identifying meteorological influences, not merely in creating the classification dataset used. A title change is warranted, and we have altered the new title to be "Identifying meteorological influences on marine low cloud mesoscale morphology using satellite classifications". As the reviewer points out, the description of the classification dataset is primarily carried out in Yuan et al. (2020), and so no emphasis on it needs to be given in the title of this manuscript. This manuscript is primarily focused on the meteorology, and so we did not spend too many words on describing the methods behind the classification or its accuracy, but we agree with the reviewer's comment (both comments (1) and (6)) that the accuracy of the deep learning and a brief description of its method and accuracy is warranted. The following language has been added to section 2.1, paragraph 1, in the description of the classification dataset:

Old text: [list of scenes] …These scenes are then used to train a convolutional neural net, which in turn is run near-globally on such MODIS oceanic scenes., and a detailed description of the classification dataset and training can be found in Yuan et al. (2020).

New text: [list of scenes]. These categories were chosen by examining the category climatologies in Muhlbauer et al. (2014) and studying regions where there was little variability in category (primarily the tropics, where disorganized MCC dominated), and identifying additional commonly occurring cloud morphologies. These (clustered and suppressed Cu) were then added to the pre-existing cloud categories, along with homogeneous stratiform category initially used in Wood and Hartmann (2006). Examples of these types can be found in Figure 2.

The scenes are then used to train a convolutional neural net (CNN). The CNN input data is the image of scene visible reflectance. A full description of the machine learning training and model evaluation can be found in Yuan et al. (2020); the main results are that average model precision evaluated on a test set was approximately 93% across all types. Open-MCC had the lowest precision, most likely because it was the lowest-frequency category. The largest source of model confusion was between disorganized MCC and clustered Cu, which is unsurprising as these categories have similar appearance.

Comment (2): Please see the first paragraph added above, where we have added some more explanation about the origins of these cloud types. Additional information on the types is found in Yuan et al. (2020) and we do not think it would be beneficial to repeat too much of that discussion in this work.

Comment (3): We have added in the appropriate citation for CERES (Doelling et al., 2013)

Comment (4): We have added in the appropriate citation for CERES (Gelaro et al., 2017)

Comment (5): We have re-ordered the figures such that they are now in the order first mentioned in the paper.

Comment (6): Addressed in response to comment (1).

Comment (7): The following line has been added to section 3.1: "Panel (a) shows the fraction of scenes covered by the dominant cloud type for that grid box" to address this omission, thank you.

Response to reviewer #2:

Comment on l101, l133, l202: By mass continuity, the horizontal divergence at 700 hPa would only give us the *local* vertical divergence dw/dz (or dω/dp in pressure coordinated) at 700 hPa, and while we do not expect very strong gradients of large-scale divergence with height in the marine lower troposphere, this value will be somewhat sensitive to the level chosen. Using $w_{700}/z_{700}$ results from the integration of dw/dz from the surface to 700mb, and so represents the mean horizontal divergence over that layer, making it more suitable as an estimate of large-scale divergence. To make this point clearer, we have amended this paragraph as follows:

"Note that this large-scale divergence is not the horizontal divergence at 700 hPa, but rather the mean divergence from the surface to the 700 hPa level; this follows from the mass continuity equation by considering a column of air from the surface (where vertical motion is 0) to 700 hPa. The terms large-scale divergence and 700 hPa subsidence are used interchangeably throughout; divergence is plotted instead of subsidence to allow for more straightforward comparison with surface divergence. As surface pressure varies with time, the second equality is only approximate."

Comment on l218: Corrected typo, thank you.

Comment on l218-220: The reviewer correctly interpreted this sentence, and are perhaps wondering if there was anything more to this point; we have added a parenthetical "(as expected)" to this sentence to indicate there is not deeper point being made.

Comment on l234: Corrected by adding reference.

Comment on S3.5: regarding alpha_q, we have added the following text: . The parameter $\alpha_q$ is a measure of how much the upper boundary layer moisture resembles the lower FT as compared to the lower boundary layer; a value of 0 would indicate a perfectly well-mixed boundary layer, while a value of 1 would indicate a perfectly decoupled boundary layer where the upper BL moisture was equivalent to the lower FT moisture.

$$\alpha_{qT} = \frac{q_T(upper\ BL) - q_T(lower\ BL)}{q_T(lower\ FT) - q_T(lower\ BL)}$$

For a given profile, the thermal inversion height is estimated using the maximum in lapse rate, with the inversion being the layer where the lapse rate deviation from a moist adiabat was >25% of maximum deviation (this was tuned to agree with a visual assessment of the inversion layer and worked well for all profiles). Upper and lower BL in the $q_T$ equation are taken as the top and bottom 25% of the BL depth,

while the lower FT is taken as the 500m above the inversion top. While this method may not be the most precise in individual more complex cumulus cases with more spatially and vertically heterogeneous moisture profiles, we use it for consistency and reproducibility. We also note that a joint histogram analysis of $\alpha_q$ vs cloud layer depth (not shown) produced consistent results to Wood and Bretherton (2004) and Park et al. (2004).

Regarding question 2 on this section, this is clarified in the text added above; the inversion used for diagnosing MBL depth is always the strongest thermal inversion, which tended to occur at the top of the remnant cloud layer in trade-like shallow Cu profiles. The result is that the boundary layers appeared highly decoupled as this layer has been subject to ample dry entrainment during the transition. There were some cases where the upper layer was thoroughly erased by dry entrainment that the diagnosed inversion was much shallower. A sentence has been added acknowledging that this method may not be ideal for some cases and justifying our use.

Regarding question 3 on this section, we reprocessed the profiles to include a surface mixed layer depth using the (near-)surface-derived LCL, consistent with Wood and Bretherton (2004). This allows for a more apples-to-apples comparison with the figures referenced by the reviewer showing model results of the same quantities.  We include the fit from Park et al, 2004 shown in all 3 previous papers (though we note that Neggers et al. used a lower gamma value). It certainly seems that our profiles are consistent with the results in Wood and Bretherton, though we do not have
 the sample size to say anything more insightful regarding the slight disagreements between the LES results and the plotted fit in de Roode et al. 2016. We have added a sentence (see above) stating the consistency with Wood and Bretherton (2004) / Park et al (2004).

Regarding the comparison to Lock (2009), we did briefly attempt to validate those results, but could not find a strong correlation between kappa and low cloud fraction. The most likely explanation for this is that in our observations, both kappa and cloud fraction are taken roughly simultaneously, whereas the simulations in Lock allow for cloud adjustment. It is also possible that kappa is sensitive to noise resulting from messy real-world profiles, and so we cannot say anything one way or another about its role in controlling cloud fraction.

Comment on Fig 2: added "Image scale is roughly 100 km across." to caption.

For the discussion of clustered vs disorganized MCC, the following text has been added to section 2.1: "The primary difference between these two types is that disorganized MCC represents a regime with cellular convection at some characteristic scale, though not obviously organized into open- or closed-cell regimes, while clustered Cu represents aggregated convection at a variety of scales within a scene. For distinguishing between these two types during manual labelling, scene large-scale context proved helpful."

Comment on Fig 5: This may be a monitor issue, though we do agree that for Figure 5, the colors are not as easy to distinguish, though as the reviewer notes it is a moot point as no disorganized MCC or

clustered Cu occurs in this scene. The colors are selected for consistency with the rest of the plots, where they do not present an issue, and so we will keep them as is.

Regarding the differing color scales between the two divergence plots, we have updated the plots so that both ASCAT and MERRA surface divergence are on the same color scale.

Comment on l342: typo corrected, thank you.

Additional changes to the manuscript:

Added code/data availability, added author contributions.